# What Digital Storytelling Means to the New Generation of Researchers

**Antonia Liguori \*, Karen Jiyun Sung, Lucy McLaughlin and Jennifer Stuttle**

School of Design and Creative Arts, Loughborough University, Loughborough LE11 3TU, UK;
j.sung@lboro.ac.uk (K.J.S.); l.mclaughlin@lboro.ac.uk (L.M.); j.stuttle@lboro.ac.uk (J.S.)
\* Correspondence: a.liguori@lboro.ac.uk

**Abstract:** A new, hybrid way of conceiving Digital Storytelling (DS) in applied research is presented here as an essential trigger to challenge, expand, and eventually re-frame the way in which DS is currently codified. The three methodological perspectives described in this paper share a common understanding of practice-as-research. They position themselves within three distinct disciplines—illustration, animation, and the creative arts in education—but have a strong commitment to interdisciplinarity. Each of them is trying to respond to a specific cultural and personal issue (e.g., sense of identity, mental health, attainment within competitive environments, etc.) and also serves as a prompt to reflect on a potential new aspect of DS as research, linked to the *how*, the *what*, and the *why* of these multiple and complementary applications. The intention of this paper is not to propose one alternative way of operating, but to inspire other researchers wanting to apply this approach in their work to constantly challenge any pre-conceived form and process, while prioritizing the democratization of knowledge production and informing their research process with co-design and participatory principles. The take-away message from these three case studies is that DS will be, in fact, embraced by the new generations of researchers as a sustainable practice all the more, as its many disruptions will generate spaces for co-creation and self-representation to emerge, and will stimulate everyone involved in the research process to challenge their own way of thinking and to go beyond what was codified by others and by their own practice too.

**Keywords:** DS as research method; applied storytelling; illustration; animation; interdisciplinary research; co-design





## 1. Introduction

This paper was conceived as a collaborative exploration of Digital Storytelling (DS) as a research method across a diverse range of disciplines within the creative arts, with the ambition to expand the conversation around applied storytelling and its multiple applications as a practice-based research methodology. It was inspired and nourished by the constant dialogue between a group of early career researchers (at different stages in their PhD journey) and a more senior colleague who has been acting over the years as their supervisor, mentor, and research companion. This methodological contribution was prompted by a shared desire to first define, and then adapt, DS as a research approach. It started from the premise that the DS form and process are never pre-defined, but must be co-designed with a diverse range of stakeholders and participants involved in the research. A number of *disruptions* to the conventional workshop-based DS model are suggested in this article, with the ambition to expand the current debate within an international community of practice recently reunited in the US for the 11th Digital Storytelling Conference. These *disruptions* are presented here at the initial stage of an iterative process as a way to prompt a collective exploration towards a more inclusive and sustainable progress of the practice, being inspired by the ethos of the practice itself. Within the context of this paper, by *disruption* we mean the adaptation of the *form*—the conventional DS workshop-based

approach—to respond to methodological questions raised by practitioners working in different disciplinary fields and wanting to apply certain elements of DS in a more flexible way. Nevertheless, this reflection on the potential use of new hybrid approaches is proposed here as a coherent response to the original principles and ethos of the practice that the Center for Digital Storytelling evolved out of the mixture of community arts practices in the 1990s (Lambert 2007). Those principles were in fact *disruptive*, whilst the form has often been emulated over the years across the globe as *conventional*.

When working across disciplines and different generations of researchers, the definition of the practice is *per se* a moment of enlightenment. The acknowledgement of existing boundaries is also useful to define standards, as well as being essential to allow diverse perspectives to be magnified.

## 2. Diverse Perspectives as *Disruptions* of Digital Storytelling as Research

The three methodological perspectives described in this paper share a common understanding of practice-as-research (Dallow 2003). They position themselves within three distinct disciplines—illustration, animation and creative arts in education—but have a strong commitment to interdisciplinarity (Zhang and Wang 2020). Each of them is trying to respond to a specific cultural and personal issue (e.g., sense of identity, mental health, attainment within competitive environments, etc.) and also serves as a prompt to reflect on a potential new aspect of DS as research, linked to the *how*, the *what*, and the *why* of these multiple and complementary applications. The reason why these three disciplines, and not others, are considered here is because the three researchers who wrote the following three sections decided to link their practice-based research to DS by joining the Storytelling Academy at Loughborough University, UK, at the beginning of their PhD studies. Despite coming from a diverse range of disciplinary backgrounds and academic journeys, they identified their research agenda within the wider spectrum of applied storytelling. This kind of bottom-up disciplinary alignment triggered an ever-expanding collective methodological conversation within this newly formed research group, based on the common *rule* of challenging each individual's assumptions and standing point.

The intention of each of the perspectives proposed here is not to suggest one alternative way of operating, but to inspire other researchers wanting to apply this approach in their work to constantly challenge any pre-conceived form and process, while prioritizing the democratization of knowledge production and informing their research process with co-design and participatory principles (Liguori et al. 2023a).

### 2.1. How: From the Story Circle to the Sketch Circle

DS in participatory research has been considered a prominent method of creating and expressing personal stories (Liguori et al. 2023b). It has also been particularly beneficial in transforming other conventionally non-participatory research approaches to be participatory, as in the case of illustration as research.

This first perspective offered here has been written by a researcher active in the field of illustration as research. It aims to expand *how* illustration as a visual communication method can be reframed as a participatory research method by embedding storytelling techniques. The context of application of this particular research consists of the visual representation of the sense of 'self' of South Koreans. As a South Korean, the researcher's methodological exploration was inspired by a key characteristic of that society, which has traditionally been a socio-centric society where the notion of 'we' versus 'them' prevails (Yang 2019). This perspective was solidified from a history of oppression, conquest, and colonization (Chen 2012; Pai 2000) where keeping the sovereignty of the nation was more imminent a necessity than that of individual identity. Now, as a prominent nation, the homogeneous people of South Korea have seen friction in keeping up with their social role while developing an independent sense of self (Watson 2012). Stories need to be told to release the tension between social and self-identities, and illustration is able to provide these realized stories as visual entities.

In illustration as research, the literature has exemplified that although recent practice-based scholars of illustration have conducted participatory data collections (Kim 2023; Vormittag 2023), the traditional illustration process has not incorporated a co-creative or participatory method as a part of its creation. The said scholars have conceived illustrators as 'additions' to a pre-existing community of storytellers rather than as 'partners' (Vormittag 2023). However, illustration as a means of visual storytelling of the self holds many possibilities as a powerful method of collecting and expressing stories, such as DS, without the barriers of linguistic proficiency or knowledge. Therefore, it was deemed appropriate to merge the process of illustration with that of the traditional DS technique to best discover the senses of self in South Koreans.

Rather than wholly adapting the DS method, the research aimed to adapt its particular strength in participatory story-sharing into the illustration process. The Story Circle, the first step in the traditional seven-step DS method (Lambert 2013), promotes an exceptional balance between self-reflection and community sharing of stories. Scott (2018) has noted that stories are not only a logical reconstruction of the past but are reflexive of the surroundings of when and to whom the stories are being told. Stories, then, reflect the arguments of Tajfel (1974) who stated that identities also adapt to accommodate where and with whom people find themselves. Therefore, both stories and identity need to balance between the independent sequences of experiences and their carefully curated representation for an audience.

Adapting to the methods of the Story Circle, this research has developed a new approach to story-sharing with illustration's visualization ability. The researcher has named the new process as the Sketch Circle. The Sketch Circle is an innovative development in participatory research in both illustration and DS that departs into a pivotal distinction from the Story Circle. The aim of the Sketch Circle is the "process of storying through a shared visual language" (Sung 2023, p. 56). John Berger noted that "Seeing comes before words" (Berger 2008, p. 1) and as such, drawing as communication predates—or is a foundation of—linguistic writing (see also: Clottes 2008). Illustration has used the native form of drawing to communicate ideas, information, and narratives through millennia. Yet, visual expression is not as native as it seems. Drawing as assets of storytelling, much like using verbal means of storytelling, is often considered a noble talent of the few. It is not uncommon to hear people express "I can't draw" by participants with non-creative backgrounds. Indeed, visual literacy is a learned capability much like writing is (Brumberger 2011). Then, the "I can't draw" remarks by participants entailed that they have not been taught in visual literacy as they did in verbal language, causing fear in the practice. However, such a fear also stems from what is considered a 'good' drawing. A good drawing for storytelling is not one which mimics the observable world, but which correctly tells a story. As long as the story is expressed, a 'stick figure'—a common standard to express the lack of drawing skills—is perfectly sufficient. In fact, drawing for storytelling as a physical practice has proved to be no different from writing a language (Cohn 2012). Then, verbal and pictorial storytelling is more interlinked than independent. If one can tell a story through writing, one can tell a story through drawing.

The Sketch Circle was aimed at introducing the participants to making or writing in visual codes as icons that make direct connections with their sense of self, so that they are equipped with a series of visual vocabulary that represents their stories. In engaging in the co-developmental process of drawing as storytelling, the main purpose of the Sketch Circle arises as an innovation made to allow the participants to tell a story and show it simultaneously. In other words, the pursuit of drawing the self itself is the new form of storytelling outcome. This marks the pivotal distinction away from the proceeding steps of DS that focus on creating a digital output after the Story Circle.

In its initial case study, six South Koreans were recruited and underwent a series of eight workshops to co-develop the process. Similar to the Story Circle, the participants initially met as groups of two to four people and proceeded to work intimately with the researcher for the latter half. The purpose of the workshops was to test and establish the

steps in merging drawing with storytelling, developing the methodology together while discovering the personal drawn stories. All participants were self-described non-artistic and were hesitant to dive into drawing for storytelling. Therefore, the participants were given brief introductions on how to draw before engaging in collective drawing. The participants continued to engage in verbal discussions and storytelling as they were drawing, engaging in participatory storytelling in two media: drawing and oral. Literat (2013) highlights how participatory drawing projects allow for "revealing a more nuanced depiction of concepts, emotions, and information in an expressive, empowering, and personally relevant manner" with a "lack of dependence on linguistic proficiency" (p. 84). The Sketch Circle, therefore, is not a complete departure from DS but an enhancement through combining the power of oral storytelling and the expressive experience of drawing.

In short, the addition of drawing in the Story Circle as the Sketch Circle is an evolutionary process of the DS technique that allows stories to be more co-creative, expressive and reflective. It is also a crucial contribution to the field of participatory illustration, as it addresses the gap between the maker and viewer of images in how to bridge the understanding of what a 'good image' is in visual storytelling. Finally, the Sketch Circle is aimed at enhancing the participants' knowledge of image-making so that future participatory research methods stand on higher ground to co-create more stories without reliance on linguistic competence or proficiency with technology. Drawing through storytelling will open up wider possibilities of *how* DS can be manifested.

### 2.2. What: Boosting Mental Welbeing through Story-Making

This second perspective offers an insightful reflection on how the subject matter, i.e., the *what*, can drive the adaptation and disruption of DS as research. It can be defined as a *responsive* methodological development in which some of the key characteristics of a particular creative practice—animation—were explored in a story-making process constantly shaped by the participants' needs. It provides an original lens to look at *slowness* as the ideal *tempo* to improve mental well-being through creativity.

There is nothing novel about the assertion that art can be a therapeutic experience for the artist (Argyle and Winship 2015; Heenan 2006). However, as demand for mental health provisions increases, further investigation of non-clinical routes to boost mental well-being and resilience is vital (Mind 2023; BMA 2023).

The research project proposed here as a case study that originated in the depths of the COVID-19 lockdowns, when discussion of mental health was growing, as many saw their own decline. It was developed under the umbrella of a wider research project called 'What's Up With Everyone?' (WUWE, whatsupwitheveryone.com accessed on 11 July 2023) which utilized storytelling and animation to improve mental health literacy amongst young people. Funded by the UK Arts and Humanities Research Council, collaborators included the Institute for Mental Health (University of Nottingham), Loughborough University, the London School of Economics, the Mental Health Foundation, Happy Space, and Aardman Animation Studios. WUWE comprised "a series of five new animated films created with and for young people about dealing with life's challenges before they impact mental health" (UK Research and Innovation 2021). At the culmination of the project, WUWE study participants stated: "a belief that the animated films had the potential to promote mental health literacy, especially for understanding mental health and reducing stigma" (Ito-Jaeger et al. 2022). While such assertions argue favourably for the efficacy of the short, animated films, what can be said for the value of the creative process itself? Specifically, the process of creating a stop-motion animation.

This specific style of animation "is captured one frame at a time, with physical objects that are moved between frames. When you play back the sequence of images rapidly, it creates the illusion of movement" (Dragonframe 2023). This is a time-consuming process, one that requires the animator to slow down and maintain focus on the task at hand. This distinctive slowness is what sets it apart from other related techniques such as film or photography. What might partaking in this experience mean for the participant who creates

the animation? What is their experience of the animation process and how might this relate to their mental well-being? Broadly, these questions are what sit at the core of this research project. Exploration of these questions began with one-day animation workshops consisting of up to five participants. It should be noted that the participants in question were not professional animators and largely had no animation experience at all. However, this need not be a barrier to partaking in the animation process. In our increasingly digital world, the technology required to create a stop-motion animation has become more readily accessible, removing the necessity for specialist equipment or extensive training (Gorman et al. 2022). Access to a smartphone or tablet subsequently allows access to a camera, and a breadth of stop-motion apps or video-editing software. Selected with the novice animator in mind, a free smartphone app [Stop-Motion Studio] was utilized in the workshops, which enabled participants to capture images and check their animation progress in real time. While all participants utilized their own smartphones, provisions were made for anyone who did not have their own technology.

At the inception of these workshops, DS was not part of the planned methodology. As the outlined research unfolded, the researcher was simultaneously exposed to DS methods employed in various other concurrent projects. Through this, it became clear that there were connections and subsequent influences of DS within the research that needed to be properly evidenced and acknowledged. Perhaps the most conspicuous was the emphasis on process over product. This was a maxim that had informed the research from the first proposal outline and one that also underpins the DS methodology (Lambert and Hessler 2018). While the creative process undertaken in either medium will almost certainly lead to an output, this is in many ways a secondary consideration. The process that is of interest is the personal one; the creative process of the animator, of the storyteller.

It was in drawing this first fundamental parallel that it became apparent how the existing DS methodology could assist and inform research design and data collection within the project, particularly following the initial animation workshops. However, this would require some dismantling of the established DS model, which at times seemed sacrilegious. These fears were assuaged by Lambert and Hessler's (2018) declaration that the DS method is there to be expanded on and adapted, alongside the already innovative work of many others. Adaptation appeared more achievable, and even encouraged. Through the subsequent process of consideration and evaluation, the practice of methodological adaptation and expansion revealed itself to be a reciprocal one. Certainly, DS could inform the animation workshop design, but it must also be considered that stop-motion animation could add a new dimension to the existing DS methodology.

Within DS, the importance of the storyteller's ownership of the narrative is paramount (Lambert and Hessler 2018). Thus proving to be a successful method of giving voice to marginalized communities (de Jager et al. 2017), the democratization of the storytelling process and the opportunity for people to reclaim their own stories is a core component of the method (Lambert and Hessler 2018). In the StoryCenter process, or what may be considered 'classical DST' (Henrickson et al. 2022), storytellers will often use a mix of media including their own footage, stock images, and text to visually support their narration. In this way, they are constrained by what can be captured realistically or by what footage is available to them at the time. If, in place of this classic approach, the storyteller employs stop-motion animation as the visual component of the story, then there is the potential for new realms of visual language to become available to them. Utilizing this medium can enable participants to "bring into view what cannot be captured by a camera—the oneiric, the absurd, the surreal" (Morelli 2021). While existing objects can be used in the animation process, participant ownership and autonomy could be further increased by crafting the objects to be animated. Within the previously discussed workshops, materials such as clay, paper, and a variety of craft supplies were supplied to participants in order to facilitate this. As such, the world in which their story appeared was entirely of their own design. Thus, it seems reasonable to consider that stop-motion animation has the potential to further

increase participant ownership of their digital story, thus expanding upon the classical DS method.

The emergence of this hypothesis through the initial study is of great interest and continues to inform the ongoing research. While DS and stop-motion animation have crossed methodological paths before (Hurtado-Mazeyra et al. 2021; Pérez et al. 2014) the potential synergy between the two surely deserves further consideration. Moreover, the potential benefits of taking a somewhat 'anarchic' approach to established methodologies could also be reinforced by this discussion, perhaps for the relief of new researchers in particular.

*2.3. Why: Questioning the Application of Storytelling and Its Sustainability in Education*

The third perspective is grounded in the field of creative arts in education and stimulates interesting questions on *why* DS should be applied to improve learning experiences and *why* it should be adapted to the specific contexts by embedding co-design techniques while developing educational resources. This case study at a first reading might seem very specific to the British educational system, but in reality, its principles are applicable to various educational contexts where STEM subjects (science, technology, engineering, and mathematics) are prioritized and competition among students is encouraged.

Over the last decade, there has been a significant reduction in the teaching of creative subjects in formal education. In recent years, in the UK, the General Certificate of Secondary Education (GCSE) specifications have been rewritten with significant alterations increasing their complexity and content. An additional change was that many English Baccalaureate (EBACC) subjects removed coursework, resulting in these now larger and more complex specifications being 100% exam assessed. Lord Baker, the former UK Minister for Education, stated during a radio interview that "If you are great at memorising facts and figures then it works for you but unfortunately if you are not great at memorising then it puts you at a disadvantage" (BBC, exam). The increased curriculum content for some non-arts subjects has resulted in less time for teachers to spend on creative teaching methodologies (Broadhead 2022) and an increased pressure on teachers and students to 'make the grade'. Ken Robinson (2006) states: "We are now running education systems where the worst things you can make are mistakes. And the result is that we are educating people out of their creative capacities". The reduction in creativity is impacting on student well-being (Davies et al. 2013) but also students' fear of failure is causing an increased level of anxiety around exams (Atkinson et al. 2019).

Research shows that the arts are proven not only to impact positively on student's mental health and well-being (Merschel 2020) but also on how they develop academically. Within primary education, there is an emphasis put on creativity and storytelling. Studies have analysed the impact of storytelling on children, proving that early use of stories within education benefits general creativity and the ability to learn (Mundy-Taylor 2013). Holderness, who was involved in the National Oracy Project states that "Children are constantly making sense of their world through stories or relating their own experiences or fears to those found in stories." (Holderness and Lalljee 1998, p. 97). Storytelling has also been used as a creative tool in higher education to help with memory and academic attainment (Eck 2006; Smeda et al. 2014; Aktas and Yurt 2017). Despite the evidence of the benefits of storytelling in higher and primary education, the research on its possible impact in current secondary education is minimal.

Storytelling is something that all humans do, often without consciously doing it (Gottschall 2013). It does not require any additional support or resources, and every teacher has stories to tell about their subject. Applied storytelling has proven to increase knowledge retention, understanding, and engagement (Eck 2006; Mundy-Taylor 2013). It is also a tool that can increase self-awareness, self-esteem, and empathy; skills that are needed to develop the 'whole student' (Mundy-Taylor 2013; Lambert 2013; Liguori 2015). Despite the research showing that this is a method that could impact positively if used in schools, the sustainability of these methods is in question. In previous studies, teachers have said that

despite the positive impact they could not continue with the storytelling methods, because they need to be delivered by a specialist storyteller or performer, or they do not meet the school's priority or curriculum, or there is no time (Heinemeyer 2020).

The research presented here involves collaborating with local schools to maximize the influence and impact of applied storytelling methods by increasing sustainability. The practice-based activities consisted of a series of focus groups and workshops with a self-selected group of sixth-form students (the final two years of secondary education, ages 16 to 18 years), following (and followed by) knowledge exchange sessions with their teachers. It is the aim that by listening to the voices of both students and teachers about the priority areas in the school and the constraints in the system, creative materials can be co-designed to allow sustainable creativity in the classroom that can be taught by any teacher regardless of subject discipline. This case study also has a focus on formatting a process to collect voices on priority areas within the school so that teachers can repeat the process in the future without a professional researcher or storyteller present. A number of lesson plans would in fact be co-designed with the students to provide the teacher with concrete examples of activities to use storytelling in the classroom without the need of additional finances or time for specialist organizations. In this way, this still-ongoing, practice-based research aims to make classroom creativity more sustainable. It could also allow for further benefits of using stories, such as being able to apply knowledge, thus developing curiosity of learning and developing students' empathy, self-awareness, and voice.

DS could be a specific tool for increasing self-esteem and resilience in students. It can be a powerful tool, not only for self-reflection and for giving people a voice that would not otherwise be heard but also to develop emotional literacy, particularly through steps one and two of the seven-step DS process; for example, "Owning your insights" and "Owning your emotions" (Lambert 2013). This approach is now being used more frequently in education, as many believe that it aids a student's self-reflection and learning, which has also been seen through the study of media production (Jamissen et al. 2017). The media production process for students allows them to develop skills in critical thinking, collaboration, creativity, and communication (Liguori 2015).

Once students have developed an awareness around their own stories, applied theatre methods could allow students to take a story and change it, or see it from a different perspective. Helen Nicholson (2014, p. 174) states: "The gift of applied theatre is that it offers an opportunity for ethical praxis which disrupts horizons, in which new insights are generated and where the familiar might be seen, embodied and represented from alternative perspectives and different points of view." The combination of both applied storytelling and theatre methods has the potential to be a powerful combination. Storytelling methods would allow for self-reflection, while theatre methods would equip students to physically embody different perspectives on stories, further empowering them to see that they are capable of altering their story or see it in a different light.

This research is also assessing the process of developing material using stories in order that this might be adapted in other schools, so that others can use applied storytelling methods for their own unique priorities or students' needs. Just as stories empower voices to be heard, schools could be empowered to use stories themselves to maximize the impact in their own work across communities of learners and teachers.

## 3. Conclusions

The three perspectives presented through the lenses of three researchers active in the field of illustration, animation, and creative arts in education provide different but complementary reflections on the disruptive power of DS as research, both in its conventional model and in its various adaptations. They focused on the *how*, *what*, and *why* as possible prompts to generate creative *disruptions* to the existing DS approach, and expand horizons with the ambition to ensure longevity for co-created and co-creative practices.

What we learn from an illustration-driven storytelling approach is that the adaptation of the Story Circle as Sketch Circle could dismantle the barriers of linguistic proficiency

or knowledge. This emerging new approach has been demonstrated to allow the self and the social to be reconciled and celebrated in the co-creative act of illustrating, by making information emerge visually. It has also reiterated the importance of differentiating representation from manifestation while shifting from an audience-driven creative process to a story-driven co-illustration process. Drawing the *self* was presented as if the process was *the* outcome, as if for stories to be expressive and reflective they should be more co-creative.

It must be emphasized that the exchange between and expansion of disciplines is mutual as demonstrated by the case study on animation, where DS was introduced at a later stage as an inspiration more than as an actual research method. The use of stop-motion animation for mental well-being, which was the core objective of this second case study, moved the focus from output to process, and this is where DS emerged for its commonality and shared values. What this perspective demonstrated very clearly is that valuing DS to inform research design and data collection is an effective thinking tool, even when applying a different creative and participatory method. The *disruption* that the second perspective suggests that it consists of the additional contribution of animation to expand the possibilities of visual representation within the conventional DS model and to visualize the invisible and the non-existent, by expanding the role of imagination in story-making.

Nevertheless, as the third case study made evident, opportunities for imagination to be nourished and to emerge are at risk. The reduction of creativity in formal education that was presented in the third perspective is, in essence, a call for action for researchers from any field to apply story-based creative approaches to generate new forms of shared knowledges (and here the plural is necessary). DS could in fact be embraced by the new generations of researchers as a sustainable practice all the more, as its many *disruptions* can generate spaces for co-creation and self-representation to emerge. In line with the DS ethos and principles, those *disruptions* could stimulate everyone involved in the (research) process to challenge their own way of thinking and go beyond what was codified by others and by their own practice too.

As the Italian piano composer and philosopher, Giovanni Allevi, recently said, "One should never be afraid to break the rules if our heart calls for it. Never be afraid to destabilize a system: the need to change is in its nature" (Allevi 2023).

Notwithstanding this, to be coherent with the social value and benefits of *disruptive* co-creative practices, by "our heart" here we allude to 'the heart': the principles and values of the collective of researchers, stakeholders, and participants involved in a shared effort to address personal and social challenges through a co-designed research practice always open to future *disruptions*.

**Author Contributions:** Conceptualization, A.L.; methodology, A.L., K.J.S., L.M. and J.S.; writing—original draft preparation, A.L., K.J.S., L.M. and J.S.; writing—review and editing, A.L., K.J.S. and L.M. All authors have read and agreed to the published version of the manuscript.

**Funding:** The WUWE project mentioned in Section 2.2 was funded by the UK Arts and Humanities Research Council (Project Reference: AH/T003804/1).

**Institutional Review Board Statement:** Case study 1 was approved by the Ethics Review Sub-Committee at Loughborough University on 29 October 2021 (Review Reference number 2021-5853-4928); Case study 2 on 8 February 2022 (Review Reference number: 2022-6292-7913); Case study 3 on 24 January 2023 (Review Reference number: 2023-11397-12669).

**Informed Consent Statement:** Informed consent was obtained from all subjects involved in the study.

**Data Availability Statement:** Data supporting reported results of the three case studies will be available on the institutional repository at Loughborough University: https://repository.lboro.ac.uk/ (accessed on 14 July 2023).

**Conflicts of Interest:** The authors declare no conflict of interest.

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
