# Peer review of "What Digital Storytelling Means to the New Generation of Researchers"

_socsci, doi:10.3390/socsci12090485_

Round 1

Reviewer 1 Report

The article's overall argument of the 3 disruptions to Digital Storytelling (DS) is very persuasive and original. Its particular strength is how it adds substantive insight and understanding about DS. 

However, the presentation of the work needs improvement. Many details of the argument are missing and which, if filled in, would significantly strengthen the robustness of the argument. 

Specifically, these details are as follows:  

1) The motivation for these disruptions is not explained fully. The authors express their "shared desire" to "...adapt and disrupt" DS, and then gesture towards some sense of co-design, inclusiveness and sustainability. But what exactly is exclusive and unsustainable about DS in its current/original form? Be very clear/describe very clearly your problem space (in the absence of an explicit research question). What is the issue you are seeking to address? 

2) How did the three disciplines - illustration, animation and creative arts - emerge? Were they just happenstances? What was the criteria for inclusion? How did the group start hanging out? Why were other approaches not considered? *Were* other approaches considered? All justifications for method and argument must be made clearly. Otherwise, it all sounds very random, and research should not be random. 

3) Section 2.1 - I am not entirely persuaded by the description of participants' grasp of visual language - ie "drawing for communication"; or establishing that "series of visual 'icons'" - for DS. Drawing to communicate narrative - cf drawing as illustration - are completely different things. If some examples - or even 1 example - could have been discussed in the section, perhaps with the drawings attached in an appendix, that would (indeed!) illustrate the point here much more clearly. I get the point about drawing as a disruption to "how". But there are many unexplained nuances: this is not just about drawing, but drawing *for storytelling* (*that* is the disruption). The details of the "for storytelling" part needs to be fleshed out much more clearly.     

4) Section 2.2 - A lot of discussion about stop motion as a general method being an additional method/dimension to DS. That's fine, but, again, it's the details:

(i) Stop motion is more than just about making objects and moving them. One needs to know how to film them (working out positioning, angles etc), then put the shots into a movie editor and string them together to make a coherent video in its visual language. There are thus various technical, skills and knowledge barriers. How would these be overcome, specifically as the authors proclaim this as being "new realms of visual language [becoming] available to [storytellers]"?    

(ii) I don't understand how stop motion can "increase participant ownership". How so? 

(iii) Again, justification: why stop motion? Why not photographs, films, videos, Tik Tok? What's so different about stop motion? 

5) Section 2.3 (erroneously numbered 2.2) - Ok, but be careful about asserting conclusions without empirical evidence. Eg "...applied theatre methods would allow students to take a story and change it...." (line 290). If you are saying "would", you are asserting that it *will* happen - that needs to be backed up by primary data of some kind, which you do not have. I get that you are projecting a possibility, which then should have been expressed as "*could* allow" rather than "would allow". Again, it is the lack of such detail and care of expression which let down the robustness of the article.  

6) There are also various careless mistakes throughout the manuscript. Examples include incorrect numbering (2.2 appears twice)); the Ken Robinson quotation (lines 242-44) is not cited with a timestamp from his talk; acronyms (even one as seemingly self-explanatory as "DS") must be spelt out at first reference etc. Please do a thorough check and proofread of your work before submission. 

The prose is generally clear and fluent. However, there are a number of long and convoluted sentences which are difficult to read and understand (for eg the second paragraph of the Intro is one whole sentence! Ditto the second paragraph of Section 2). Some crisp editing would really improve this work.

There are also some grammatical errors, eg "Being the researcher a South Korean 67 citizen" (line 67-68). Terms of art are also inconsistently capitalised, eg sometimes it's "Illustration" and sometimes it's "illustration". Thorough proof-reading would improve this work enormously.  

Author Response

We are grateful for your supportive, detailed and very useful comments.

You can find our responses below (point by point highlighted in green) and see our edits in track-changes in the revised version of our paper in attachment.

  • The motivation for these disruptions is not explained fully. The authors express their "shared desire" to "...adapt and disrupt" DS, and then gesture towards some sense of co-design, inclusiveness and sustainability. But what exactly is exclusive and unsustainable about DS in its current/original form? Be very clear/describe very clearly your problem space (in the absence of an explicit research question). What is the issue you are seeking to address? 

The introduction was revised and expanded to clarify this aspect.

  • How did the three disciplines - illustration, animation and creative arts - emerge? Were they just happenstances? What was the criteria for inclusion? How did the group start hanging out? Why were other approaches not considered? *Were* other approaches considered? All justifications for method and argument must be made clearly. Otherwise, it all sounds very random, and research should not be random. 

Explained in section 2.

  • Section 2.1 - I am not entirely persuaded by the description of participants' grasp of visual language - ie "drawing for communication"; or establishing that "series of visual 'icons'" - for DS. Drawing to communicate narrative - cf drawing as illustration - are completely different things. If some examples - or even 1 example - could have been discussed in the section, perhaps with the drawings attached in an appendix, that would (indeed!) illustrate the point here much more clearly. I get the point about drawing as a disruption to "how". But there are many unexplained nuances: this is not just about drawing, but drawing *for storytelling* (*that* is the disruption). The details of the "for storytelling" part needs to be fleshed out much more clearly.     

Section 2.1 partially re-written to integrate and clarify those aspects – please, see track-changes.

4) Section 2.2 - A lot of discussion about stop motion as a general method being an additional method/dimension to DS. That's fine, but, again, it's the details:

Please find below our responses below and new/edited text in track-changes in our edited article:
(i) As would also be expected in traditional DS settings, participants will have varying levels of technical knowledge and skill when participating in stop motion. Such barriers are overcome by the use of a simple and free phone app specifically designed for creating a stop-motion animation from start to finish. Participants are given a tutorial on its use and have time to practice with the method. The complexity of the animation itself and objects used is up to the individual participant. In some cases, this very much equated to making objects and moving them but effective animations were developed nonetheless.
(ii) Participant ownership of the digital story could be increased as the participant has the opportunity to create all the individual visual components. This adds another layer of creativity and personalisation to the process. These self-made visuals would be in place of what might have previously been stock footage or images. The participant need not rely on what already exists to illustrate their personal narratives.

(iii) Similarly, this opportunity to physically craft a world in which to tell your story sets stop-motion apart from other mediums. It also lends itself well to the reflective process due to its slow nature.

  • Section 2.3 (erroneously numbered 2.2) - Ok, but be careful about asserting conclusions without empirical evidence. Eg "...applied theatre methods would allow students to take a story and change it...." (line 290). If you are saying "would", you are asserting that it *will* happen - that needs to be backed up by primary data of some kind, which you do not have. I get that you are projecting a possibility, which then should have been expressed as "*could* allow" rather than "would allow". Again, it is the lack of such detail and care of expression which let down the robustness of the article.  

Edited as suggested

  • There are also various careless mistakes throughout the manuscript. Examples include incorrect numbering (2.2 appears twice)); the Ken Robinson quotation (lines 242-44) is not cited with a timestamp from his talk; acronyms (even one as seemingly self-explanatory as "DS") must be spelt out at first reference etc. Please do a thorough check and proofread of your work before submission. 

Done

Comments on the Quality of English Language

The prose is generally clear and fluent. However, there are a number of long and convoluted sentences which are difficult to read and understand (for eg the second paragraph of the Intro is one whole sentence! Ditto the second paragraph of Section 2). Some crisp editing would really improve this work.

Checked

***

Warmest regards!

Reviewer 2 Report

The paper socsci-2534509 entitled "What Digital Storytelling means to the new generation of researchers" covers the very interesting topic of Digital Storytelling as a research approach in various artistic disciplines.

The author covers three methodological perspectives addressing aspects such as the representation of identity in South Korean society, mental well-being through the process of creating stop motion animations and the application of storytelling to education. However, it does not achieve a clear and precise objective with the study by not providing content of relevance to the scientific community.

Previous studies are mentioned in the review of related literature that can be improved and focused on the subject matter of the study, thus providing more context to the article.

The methodology presented in the article is quite brief and needs to increase the rigor of the research, allowing other researchers to replicate this methodology in order to apply it to other fields or disciplines. A more detailed description of how the study was carried out, the data collection methodology, if any, as well as the specific procedures, is required.

The study focuses on South Korean society, mental wellbeing through StopMotion which the author himself indicates is not the focus of the current thesis creating confusion as well as the application of storytelling to education focusing on the UK education system.

There is no relationship or link between these three aspects, which are completely disconnected.

As for the results and conclusions, it would be of great value to include the results and analysis of each case study in order to have more solid conclusions that contribute to the scientific community in a relevant way.

I have some doubts about the concept of "Sketch circle", which lacks sufficient bibliography with only one reference.

So I am afraid I do not see enough originality here to publish this version of the article. Modifications are needed to address the comments made.

Author Response

We are grateful for taking the time to carefully read our paper and send your comments. We have considered all your suggestions (and the very useful and supportive comments received from the other two reviewers) and edited our paper accordingly. We believe that is now a stronger and more convincing piece.

You can find all our edits in track changes in the file in attachment.

Warm regards!

Reviewer 3 Report

A brief Summary

This article explores new DS methods as used in three case studies to inclusively engage storytelling practices further – to define and disrupt DS as a research approach that is not predesigned and should be co-designed with participants.

These three case studies focus on the how, what and why, using various new storytelling processes – illustrations, animation and education, with the process itself providing the impact – in these examples, a sense of identity, mental health, and achievement within the competitive teaching and learning environments of a contemporary classroom.

The authors are calling for disruption within Lambert’s traditional DS process model in order to not only collaboratively co-create Dstories, but to also pay close attention to the impact and value the actual process within the DS framework can offer.

These case studies are not suggesting these are the only new ways of tweaking traditional DS practices, but calling on researchers across disciplines to playfully adopt and adapt the processes of expressing stories to ensure inclusive storytelling and the act of sharing co-created stories has powerful impact and unlimited possibilities to engage, heal, and learn. And if the expectations of impact are reframed by researchers to also include examining the process as an impactful outcome in its own right, this opens more possibilities for DS and a research method to use.

General concept comments

Article:

The article raises interesting concepts of how DS are made and categorised, drawing attention to the multiple ways stories can be told, with unique methods adopted to overcome barriers, and usefully engage participants to co-create stories – of self through illustration techniques; safely sharing and presenting mental health concerns via stop motion animation; and as a way of improving learning experiences using Creative Art education practices.

Review:

The review is an interestingly provocation, suited to the journal theme and an article this reviewer very much enjoyed reading. It highlighted what many storytelling researchers tacitly know – that is that the process is where the magic happens, and the final output is only the product of the exercise, and I commend the authors for taking a broad approach to recognising and illustrating three possibilities for uses in other disciplines, such and education and health.

I would however encourage the authors to provide more detail of each case study and include real examples of the participants engaging with illustrating, animating, or roleplaying their stories, so that the reader can also perceive how the authors’ assumptions were extrapolated to form valuable shared new knowledge.  

Specific comments:

       The manuscript is broadly clear, relevant to the field and presented in a structured manner.

       The cited references are generally an even mix of recent and older publications, and there are a couple of citations that are not listed in the references, or the link in the reference is broken (see table of comments).

       The manuscript is scientifically sound in its premise, and I look forward to understanding further the background and context in which each of the case studies were carried out.

o   More information for instance about the workshop participants and facilitators would be helpful, such as:

§   was the aim of each workshop to disrupt the process or was this a secondary output realised in the subsequent examination of the workshops?

§   What were the number of participants?

§  One project was funded through the Arts and Humanities Research Council and this link does not work to view detail. It might be a good idea to add an acknowledgement section re: funding, to acknowledge participants and facilitators and to detail projects more specifically

o   More statistical data would be helpful

The article is novel, and thought-provoking, but would benefit from offering more detail about the three case studies.

The work fits the journal scope and theme.

The quality of the article is good and would only require minor editing and tightening of some sentences (I have made some suggestions below). The layout and structure are good, and the English language proficiency is appropriate.

I commend the authors for their work and collaboration, and in highlighting case studies from different geographical locations and cultures, making the article internationally engaging and of relevance to readers across many disciplines.  The work is worth publishing as it does advance current thinking in DS, and as mentioned would stimulate conversations with a wide range of readers for other fields who I hope will see the potential benefit of interdisciplinary partnering with DS researchers to apply DS research processes in other disciplines.

        comment

line number

not sure it is clear what you mean by...’disruption being inclusive and sustainable progress’- can you clarify simply?

35

Suggest rewrite and make clearer this long sentence/para.

43

consider rewriting sentence for clarity

67

Maybe simply start this sentence, ‘As a South Korean researcher’…

69

Perhaps, ‘This perspective was…’

70

‘The said scholars…’ suggest making sentence clearer

80/81

is Story Sketching democratic and inclusive if you need illustrating skills to participate? Or is the participant directing the artist in what they want drawn- and this is what you are calling co-creative storytelling?

115

I found this interesting and wanted to understand the process and context more- how did this session work- who were the participants etc? was it run in community or just with other ECRs?

119

Again, I was thinking, how is drawing co-creative...does more than one person draw...or does the illustrator/researcher draw the story they are hearing from participants? Can you explain/give an example from the participants actions

133

but need proficiency in drawing?

138

this sentence perhaps needs clarifying

147

The link to this study does not work

162

Arguably you still need skills to do this type of animating?

207

Do you have a reference for this claim you can add in?

244

Within this para there a 3 older references…do you have more recent ones to use?

238-54

The reader needs more context/detail of case study here -to know who is listening to whose voices?

269-270

Can you please give an example of how this is possible?

278

Suggest rewriting this sentence for clarity

302

Perhaps dismantles barriers of linguistic proficiency, but perhaps requires artistic proficiency? If not, why not? Again, a little more detail/explanation from case studies perhaps

313

This quote does not appear in the references, so add it in

343

good- small edits in comment section

Author Response

We are grateful for your thoughtful, detailed and very useful comments. We have tried to carefully respond to your suggestions and improve our paper. We believe that is now a stronger and more convincing piece.

You can see all our edits as track-changes in the revised version of the article here in attachment and our responses to your detailed suggestions in the table below.

        comment

not sure it is clear what you mean by...’disruption being inclusive and sustainable progress’- can you clarify simply?

Clarified, please see edited text in track changes

35

Suggest rewrite and make clearer this long sentence/para.

Done

43

consider rewriting sentence for clarity

Done

67

Maybe simply start this sentence, ‘As a South Korean researcher’…

Done

69

Perhaps, ‘This perspective was…’

Done

70

‘The said scholars…’ suggest making sentence clearer

Done

80/81

is Story Sketching democratic and inclusive if you need illustrating skills to participate? Or is the participant directing the artist in what they want drawn- and this is what you are calling co-creative storytelling?

Process and definition clarified – please, see track changes

115

I found this interesting and wanted to understand the process and context more- how did this session work- who were the participants etc? was it run in community or just with other ECRs?

Process and definition clarified – please, see track changes

119

Again, I was thinking, how is drawing co-creative...does more than one person draw...or does the illustrator/researcher draw the story they are hearing from participants? Can you explain/give an example from the participants actions

Clarified, please, see track changes

133

but need proficiency in drawing?

Clarified, please, see track changes

138

this sentence perhaps needs clarifying

done

147

The link to this study does not work

The linked AHRC blog has since been absorbed by the UKRI site. The reference has been updated: Academics join forces with Aardman to tackle mental health crisis – UKRI

162

Arguably you still need skills to do this type of animating?

Yes, there is a certain level of skill associated with animation but I would argue this is somewhat relative to the expected output. As would also be common in traditional DS settings, participants will have varying levels of technical knowledge and skill when participating in stop motion workshops. Such barriers are overcome by the use of a simple and free phone app specifically designed for creating a stop-motion animation from start to finish. Participants are given a tutorial on its use and have time to practice with the method. The complexity of the animation itself and objects used is up to the individual participant. By providing participants with simple and effective tools and materials, the medium becomes very accessible. This is compounded by the focus on process over product, reducing pressure or expectations on participants.

207

Do you have a reference for this claim you can add in?

Added

244

Within this para there a 3 older references…do you have more recent ones to use?

Added

238-54

The reader needs more context/detail of case study here -to know who is listening to whose voices?

Added

269-270

Suggest rewriting this sentence for clarity

Done

302

Perhaps dismantles barriers of linguistic proficiency, but perhaps requires artistic proficiency? If not, why not? Again, a little more detail/explanation from case studies perhaps

More details added in the case study

313

This quote does not appear in the references, so add it in

Added

343

Warmest regards and thanks again!

Round 2

Reviewer 2 Report

The document socsci-2534509 whose title is "What Digital Storytelling means to the new generation of researchers" covers a very interesting topic such as Digital Storytelling as a research approach in various artistic disciplines. In this latest version of the document, the author has mostly attended to the modifications and issues raised by the reviewers. The document is better structured and clearer, and each of the perspectives has been addressed in greater depth and detail. The writing of the text is more fluid and coherent, which makes it easier to read, and the bibliographical references have improved in quality. The connection between the sections is better justified and the results and conclusions provided can offer future researchers interesting lines of research. Although there are some aspects to improve, I consider that the document can be approved for publication.